# LEARNING TO INFER RUN-TIME INVARIANTS FROM SOURCE CODE

**Vincent J. Hellendoorn**
Carnegie Mellon University
5000 Forbes Ave
Pittsburgh, PA 15238, USA
`vhellendoorn@cmu.edu`

**Premkumar T. Devanbu**
UC Davis
1 Shields Ave
Davis, CA 95616, USA
`devanbu@ucdavis.edu`

**Oleksandr Polozov & Mark Marron**
Microsoft Research
14820 NE 36th Street
Redmond, WA, 98052
`{polozov,marron}@microsoft.com`

## ABSTRACT

Source code is meant to be executed, as well as read. Developers reason about its run-time properties by inferring "invariants", which constrain program behavior; but they rarely encode these explicitly, so machine-learning methods don't have much aligned data to learn from. We propose an approach that adapts cues within existing if-statements regarding *explicit* run-time expectations to generate aligned datasets of code and *implicit* invariants. We also propose a contrastive loss to inhibit generation of illogical invariants. Our model learns to infer a wide vocabulary of invariants for arbitrary code, which can be used to detect and repair real bugs. This is complementary to trace-based methods, such as Daikon. Our results confirm that neural models can learn run-time expectations directly from code.

## 1 INTRODUCTION

Software developers make many inferences while reading code, about its performance, correctness, and run-time behavior. The latter is often summarized by *program invariants* – generic assumptions about code behavior, e.g. that a list index never escapes its bounds. Automatically inferring these is hard: predicting run-time behavior is undecidable, thus sound analyses are limited to small programs such as simple loops (Sharma et al., 2013a; Padhi et al., 2016). Ernst et al. (2007) pioneered learning them from highly informative *execution trace data*, but it requires access to realistic program inputs. Developers also rarely "assert" invariants explicitly, so static learning data is limited.

Yet these obstacles may be largely artificial. For one, practical programs rarely take on an exponential range of values, and developers reliably infer run-time constraints from a local context, using their past experience and cues from the code itself. What's more, developers do encode many run-time constraints in their code, in the form of if-statements. These guard blocks of code that should only execute under a given condition, such as that a map contains an element, or a parameter is not null. Interestingly, programmers rarely guard all possible conditions (e.g. nullity of all parameters), often rather relying on other (calling or called) methods to ensure them.

Our central claim is that the natural distribution of programs includes many groups of similar functions, some of which assert run-time assumptions explicitly, and with much detail, while others vary along these dimensions. If true, we can use *explicit* conditional checks that guard blocks in functions to teach our models about the *implicit* invariants of unguarded blocks in similar functions. Furthermore, we conjecture that in such comparable samples, the condition is both *salient* (since it is checked explicitly) and *natural* (since it is written by humans). Learning from such examples is thus a very appropriate training signal for inferring practically useful invariants.

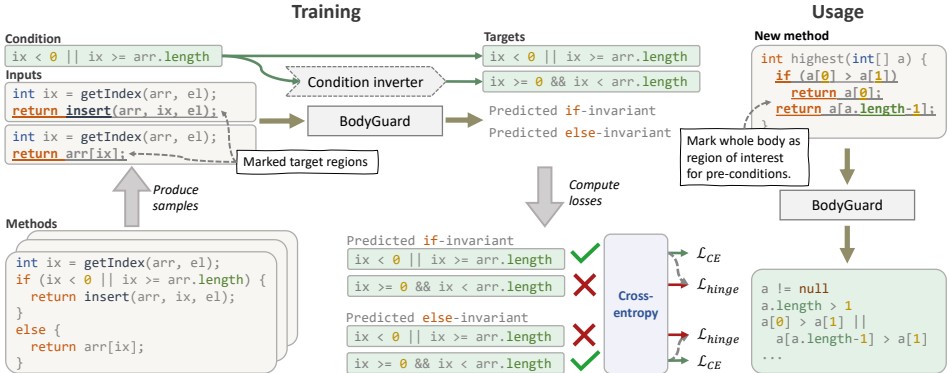

Figure 1: An overview of our learning approach. We extract samples from if-statements in Java methods by removing the guard and assigning it (or its negation, for the else block) as the target invariant(s) of the previously-guarded block. We train using the cross-entropy, plus a contrastive loss of this entropy relative to that of its logical negation (per sample).

Figure 1 shows how we exploit this symmetry between explicitly and implicitly guarded code: we remove explicit guards to generate the former. By removing these one at the time and filtering out implausible programs, we generate millions of realistic samples of functions and (now-)implicit run-time constraints. Our model, BODYGUARD, predicts a rich vocabulary of conditions about arbitrary code from new projects, and can be used to find & fix real missing-guard bugs with over 69% (repair) precision at 10% inspection cost. It also predicts more than two-thirds of Daikon's invariants that could previously only be inferred with run-time data, and some entirely new ones that can be validated automatically with trace data. This is a significant next step in learned static analysis, being the first to reliably produce natural invariants from arbitrary code alone. More generally, we show that learned models can implicitly represent behavioral semantics, just from code.

## 2 APPROACH

We train our model using a new loss function on a large corpus of open source data and evaluate it on two downstream tasks. For conciseness, we provide a high-level overview of our approach here.

### 2.1 DATASETS

We generate samples following Figure 1 from top-starred Java projects from Github, which we split by organization into training (920 projects), held-out (19 projects), and test data (61 projects). We extract all methods from each Java file and generate one sample for each (side-effect free) if-statement by removing the guard and storing its condition. As Figure 1 shows, this produces one or two equivalent code fragments for which the statement's condition (or its negation) is a precondition. The resultant sample contains the entire method (minus conditional check) as context, with the *range* of tokens where the invariant condition applies indicated.

Our model generates run-time conditions for an indicated range of code in a method. We evaluate it in two settings. First, we collect real missing if-guard bugs for our model to repair from 10K Java Github projects, by parsing their (∼8M) commits for ones that a) introduced exactly one if-guard, and b) are described as bug-fixing (Ray et al., 2016). We find ca. 3K of these. Second, we use Daikon (Ernst et al., 2007) to collect execution trace data from a smaller set of eight projects that we manually instrumented, to compare our predictions to both Daikon's own and to traces directly. This helps us both assess the validity of our invariants on real executions, and, more generally, understand the *inference gap* between static and dynamic information for source code analysis; i.e., is run-time data (when present) strictly more useful than code, or are the two information sources orthogonal?

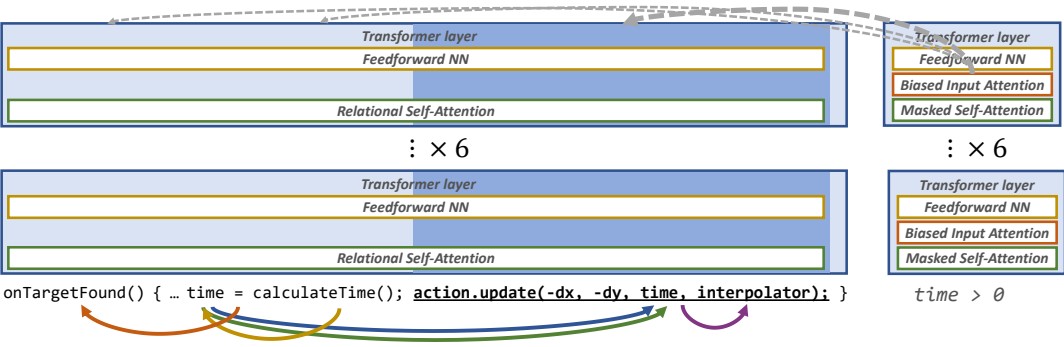

Relations: parent, computed-from, def-use, next-use, next-token

Figure 2: Schematic overview of our model: an 8-layer Transformer encoder/decoder, augmented with relational data (Hellendoorn et al., 2020), and a decoder bias towards the invariant's range.

## 2.2 MODEL SETUP

BODYGUARD uses a Transformer encoder-decoder model (Vaswani et al., 2017) to translate methods to invariants (see Figure 2). Syntactic & semantic code metadata, e.g. the AST and data-flow relations are known to help code understanding (Allamanis et al., 2018), thus we incorporate them using the relational attention mechanism from Hellendoorn et al. (2020). It adds a relation-specific bias term $b_{ij}^r$ into the query-key comparison of scaled dot-product attention: $e_{ij} = (\mathbf{q_i} + b_{ij}^r)\mathbf{k_j}^\top/\sqrt{N}$. The (learned) bias is sensitive to known relations $r$ between tokens $i$ and $j$ (if any, and summed if multiple), to allow the model to selectively sharpen (or dampen) the significance of each relation. With 512-dimensional states, 8 heads, and 8 layers on both sides, our model has $\sim$67M parameters.

We created our own program graph extractor for Java using the Eclipse JDT parser. In following with prior workAllamanis et al. (2018), we extract five commonly used kinds of edges that represent different forms of relations in code. This includes lexical dependencies ("next-token" and its reverse, "prev-token"), syntactic (AST) relations ("parent"/"child"), and various data-flow relations ("last-use", "next-token", and "computed-from"). As is common, for each of these relations we include a separate edge for its reverse, yielding 10 edge types total.

Finally, we use the same "leaves-only" representation as Hellendoorn et al. (2020) to limit our inputs to tokens only, by rerouting edges that connect non-terminal nodes to their representative syntax token (e.g. from an if-statement node to its "if" code token). We reuse this relational mechanism between the decoder and encoder, in order to produce only invariants for the relevant range of tokens (using a unary relation, i.e., "is part of range") Ding et al. (2020).

## 2.3 DECODING LOGICAL STATEMENTS

In conditions, small syntactic differences lead to drastic changes in run-time behavior. Removing esp. `if`-`else` statements as we do naturally yields code fragments with very similar, but logically opposite (e.g. '`!= null`' vs. '`== null`') conditions. We supervise our model to encourage its representations for *syntactically close* but *semantically opposite* statements to be distinct by introducing a *contrastive hinge loss* term. For every training sample, we also decode the invariant's logical negation, but require it to have a higher entropy than the real target. Concretely, given a statement $inv$ with tokens $t_i$ and a negating function neg, we use the cross-entropy loss $\mathcal{L}_{\text{CE}}$:

$$\mathcal{L}_{\text{CE}}(inv) = -\sum\nolimits_{i=1}^{|inv|} \log \Pr(t_i \mid t_1 \cdots t_{i-1}, context) \tag{1}$$

to compute the entropy distance w.r.t. its negation:

$$\Delta_{inv} = \mathcal{L}_{\text{CE}}(\mathsf{neg}(inv)) - \mathcal{L}_{\text{CE}}(inv) \qquad \mathcal{L}_{\text{hinge}}(inv) = \max\left(0, \Delta_{inv} - \epsilon\right)^2 \tag{2}$$

in which $\epsilon$ is the minimum desired entropy "distance" in bits. In this work, we set $\epsilon = 2$. For this *hinge-loss model*, as we will call it in the rest of this paper, we train with a loss equal to $\mathcal{L}_{\text{CE}} + \mathcal{L}_{\text{hinge}}$.

## 3 ANALYSIS

We sample our two models' held-out performance every 100,000 training steps, which produces the learning curves in Figure 3a. The hinge loss model saturates slower, as it faces the more challenging

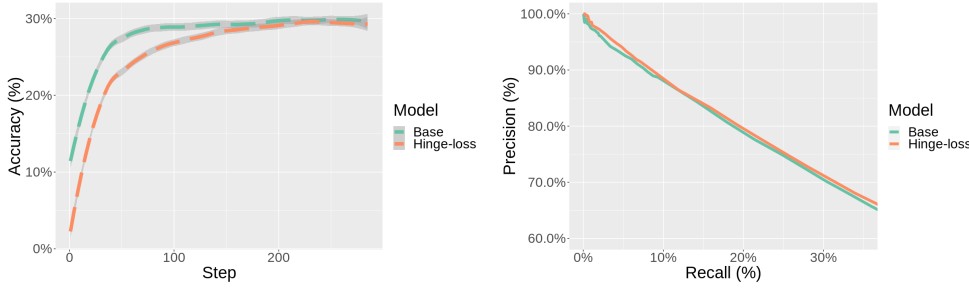

(a) Accuracy on held-out data during training.

(b) Precision-Recall response of the trained models to limits on the entropy of generated invariants.

Figure 3: Model performance during and after training, focusing on the high-precision/low-recall domain for the second (overall test accuracies: 33.8% base, 34.7% hinge-loss).

task of discriminating similar statements. However, after ca. one week of training, both models converge to approximately the same quality. It speaks to the challenge of this strategically important task that the models reach just ∼30% accuracy, due in part to the diverse vocabulary of invariants.

Figure 3b shows the evaluation on the test data, namely precision/recall behavior of the two models when ranking their predictions (from beam search) by entropy. Both models converge to (near) perfect precision at a commensurate expense of recall (breaking 80% precision around 20% recall). The hinge-loss model has a decidedly better precision/recall tradeoff, as well as higher overall accuracy.

We manually analyzed a number of these functions and our models' prediction on them, to determine the source of BODYGUARD's inaccuracies. In short, some of its mistakes are clearly a matter of modeling capacity: the vocabulary of invariants and code contexts is incredibly diverse, and our model ($<$ 100M parameters) and dataset ($<$ 1B tokens) may be too small to fully capture this distribution. Secondly, the *scope* of our samples poses a substantial challenge: we focus on modeling at the function level only, as even large functions (spanning in the order of 500 tokens) pose a challenge to the Transformer-type models that we use in terms of data usage. However, many invariants require inference at the level of files or even across function calls. Without that context, the task is inherently ambiguous for those samples. Further research on expanding our modeling horizon in software engineering is thus needed.

## 3.1 MISSING IF DETECTION

We apply our models to predicting missing if-guards, first given a localized buggy segment (top row of Table 1). This most directly relates to our training signal, where we provided the location of the guarded code. Our model achieves 29.3% accuracy (base model: 26.8%), with a favorable 69.1% precision at 10% – enough to fix 215 out of 311 bugs once located. It is likely beneficial that this task is roughly as challenging as our test one: automatically synthesized training data is often

Table 1: Bug-detection and repair results on finding and predicting missing if-guards, across two settings: given the correct location, and across all possible locations, further analyzed by aspects of the top prediction.

| Objective | Accuracy | Top-5 Acc. | Precision @10% Recall |
|---|---|---|---|
| *Location given* | 29.3% | 41.9% | 69.1% |
| *All Locations* | 10.4% | 18.8% | 39.9% |
| invariant correct | 15.2% | 24.2% | 48.1% |
| position correct | 19.4% | 43.8% | 100.0% |

overly easy compared to real tasks, which harms generalization (Hellendoorn et al., 2019b). The missing condition in these samples is (arguably) the most *salient* invariant in the entire method, not just the indicated code block, which our model should be able to prioritize. The next three rows of Table 1 show the results of running our invariant generator on every contiguous segment (up to 5 blocks) of code in each buggy method, ranking the top invariants across segments for inspection. This is substantially harder than the previous task, reducing the overall accuracy threefold and roughly halving precision. Nevertheless, that is still much better than might be expected if BODYGUARD had no location-sensitivity: we test over 30 blocks per method on average.

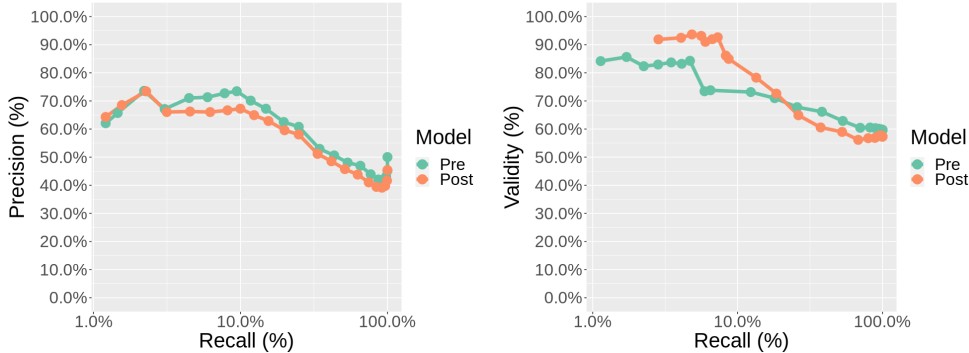

(a) Overlap between our invariants and Daikon's, on pre- and post-conditions.

(b) Overall validity in relation to varying entropy thresholds; pre- and post-conditions.

Figure 4: Results of the overlap and validity analysis of our invariants based on Daikon-extracted trace data. Note the log-scaling on the x-axis.

## 3.2 VALIDITY AND OVERLAP WITH DAIKON

Many of our emitted invariants are valid statements, even if not the real missing condition. We analyze this validity on methods for which we have trace data from Daikon (Ernst et al., 2007): as Figure 4a shows, our tool retrieves more than half of Daikon's invariants, and over two-thirds at 10% recall, from static code alone. In addition, many of our pre- and post-condition[1] are not generated by Daikon (those out of its vocabulary, or with too few observations), even at a low entropy threshold.

We assess these using a simple logical engine that compares various kinds of our invariants (e.g. collection inclusion, array length, nullity) against the recorded traces. We can check ca. 40% (12K) of our invariants in this way; on these, we find that our invariants are valid ca. 60% of the time at full recall, and this ratio increases greatly as we sharpen the entropy threshold, to over 80%, at recall values under 10%, as shown in Figure 4b.

Many of our validated invariants were not produced by Daikon, implying that static and dynamic data are orthogonal for this task. We manually analyzed 708 pre-conditions that BODYGUARD generated at an entropy of ≤0.1; 540 of these could be checked with trace data, (449 valid, 91 invalid), and 122 of the remaining 168 were found valid on inspection. In effect, over 80% of our invariants at this recall level (3.5%) are correct, and more than two-thirds of the invalid remainder could be ruled out using trace data, if available, leaving a false positive rate of just 6.5% (46/708) when execution data is available (while also adding about 200 valid invariants to Daikon's predictions). This confirms that our tool is largely synergistic with dynamic, trace-based invariant generators.

## 4 CONCLUSION

Formal invariant inference with oracles or solvers (Sharma et al., 2013a;b; Padhi et al., 2016; Pham et al., 2017) does not scale beyond simple arithmetic programs. ML-based inference (Si et al., 2018; Hellendoorn et al., 2019a; Brockschmidt et al., 2017) scales better but still usually predicts invariant validity rather than generates them. In contrast, our work makes no assumptions about the code other than the availability of a parser, and is trained specifically to generate novel, likely invariants. It succeeds thanks to the observation that typically used invariants are in a sense *natural*, like many other aspects of programs (Hindle et al., 2012; Barr et al., 2013; Tsimpourlas et al., 2020), and therefore predictable from code alone, intentionally standardized for ease of reading and writing Casalnuovo et al. (2019). Previous approaches may be symbiotic with ours: trace data or SMT solvers can be used to filter invalid invariants, and possibly guide our invariant generation. Our

---

[1]Due to the nature of our training, our tool does not quite have a proper notion of pre- or post-conditions *per se*. The best approximation we found was to mark the entire method body for the former, and just the `return` statement for the latter; there is no straightforward extension to `void` methods or ones with multiple returns, so we omit these. Neither of these methods are quite exact, so some of our predicted invariants are valid, but not pre/post-conditions. We erred on the side of caution and marked those invalid.

approach can also support other code understanding tools that struggle to navigate an exponentially large program space.

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
