# OpenReview forum: "Learning to Infer Run-Time Invariants from Source code"
_NeurIPS.cc/2020/Workshop/CAP — NeurIPS 2020 CAP Workshop_

### Official Review · AnonReviewer1 · 2020-10-29
**Deep Learning Method for Static Invariant Inference**

**Rating:** 8
**Confidence:** 4

**Review:**

====== Summary =====
This paper presents a deep learning approach to the problem of learning program invariants. The problem of invariant inference has been long studied in the PL/SE community and remains unsolved. A difficulty with a learning-based approach is that few correct invariants are written, and thus, available for training. The core idea of this paper is to use the conditions in if statements as "invariants" (pre-conditions) for the code contained in the if statement. They train a transformer encoder/decoder on annotated tokenized code. In addition to regular cross-entropy loss,  the model uses a contrastive loss to encourage the model to separate syntactically similar but semantically opposite invariants


====== Pros/Cons ======

Pros
+ good idea to try and use if statements to learn invariants
+ interesting idea to try to specifically distance negations from the invariants
+ well-written paper, easy to understand

Cons
- numerical results are not so good
- approach only works for pre-conditions, not post-condition invariants

====== Review ======
This paper targets a very difficult problem, and unsolved, in the programming languages and software engineering community. The approach proposed is totally static (does not require running the source code). The use of a contrastive loss in this space is quite novel and intuitively interesting to me, although the results aren't huge. They particularly use if-guards that were regarded as missing, that is, those which have to be there to ensure correct behavior under certain argument circumstances. This makes a lot more sense than just choosing all if statements in a program, many of which may not be about the direct arguments to the function. I wonder if even in these bug fixing commits, there were some non-invariant-like conditions introduced?

The model they propose is able to learn invariants to at least some extent, reaching around 30% accuracy. On the bright side, the model is probably not overfitting (given it has 67M parameters and there are 3K if-guards in the training set).  The precision-recall response is actually better than it looks on first glance; Figure 3(b) bottoms out at 60% precision. There is also an evaluation with the classic invariant-detection tool Daikon. The proposed tool's invariants overlap with Daikon to a large extent, suggesting that the invariants proposed are reasonable (i.e. using relevant variable names).

The paper is well-written and easy to understand.

The one downside of the paper is that the accuracy is not so high. But I think it would certainly provoke discussion, and there is a lot of room to spur further research. The use of the contrastive loss is particularly interesting to me in this space. In the long term, its effects are not too pronounced: perhaps this is because for more complex conditions, the negations are indeed quite semantically dissimilar? (e.g. in Figure 1, several characters differ). Overall, I recommend this paper for acceptance.

======= Questions ======
- How were the 5 semantic edge types chosen?
- How are post-condition invariants trained? My understanding of the technique was that it was suitable to infer pre-condition invariants, since you are only training on the if-guards to blocks. However, Section 3.2 says that "many of our post-conditions are not generated by Daikon".
- What is the main source of inaccuracy? Does it come from an inability to express the source code semantics, or to a limited vocabulary of invariants? Both? Unknown?
- Even though the code comes from different projects, do we know whether any snippets from the test data were exactly shared with the training data (e.g. what if two projects both copied code from some external library).

---

### Decision · Program_Chairs · 2020-11-02

**Decision:**

Accept

**Comment:**

As the review is positive, I recommend acceptance.